# Incorporating a Hyperspectral Direct-Diffuse Pyranometer in an Above-Water Reflectance Algorithm

**Thomas M. Jordan** [1,*] , **Stefan G. H. Simis** [1] , **Philipp M. M. Grötsch** [2] and **John Wood** [3]

1   Plymouth Marine Laboratory, Prospect Place, Plymouth PL1 3DH, UK; stsi@pml.ac.uk
2   Gybe, Portland, OR 97210, USA; philipp@gybe.eco
3   Peak Design Ltd., Sunnybank House, Wensley Road, Winter, Derbys DE4 2DH, UK; john@peakdesign.co.uk
*   Correspondence: tjor@pml.ac.uk

**Abstract:** In situ hyperspectral remote-sensing reflectance ($R_{rs}(\lambda)$) is used to derive water quality products and perform autonomous monitoring of aquatic ecosystems. Conventionally, above-water $R_{rs}(\lambda)$ is estimated from three spectroradiometers which measure downwelling planar irradiance ($E_d(\lambda)$), sky radiance ($L_s(\lambda)$), and total upwelling radiance ($L_t(\lambda)$), with a scaling of $L_s(\lambda)/E_d(\lambda)$ used to correct for surface-reflected radiance. Here, we incorporate direct and diffuse irradiance, ($E_{dd}(\lambda)$) and $E_{ds}(\lambda)$), from a hyperspectral pyranometer (HSP) in an $R_{rs}(\lambda)$ processing algorithm from a solar-tracking radiometry platform (So-Rad). HSP measurements of sun and sky glint (scaled $E_{dd}(\lambda)/E_d(\lambda)$ and $E_{ds}(\lambda)/E_d(\lambda)$) replace model-optimized terms in the 3C (three-glint component) $R_{rs}(\lambda)$ algorithm, which estimates $R_{rs}(\lambda)$ via spectral optimization of modelled atmospheric and water properties with respect to measured radiometric quantities. We refer to the HSP-enabled method as DD (direct-diffuse) and compare differences in $R_{rs}(\lambda)$ and $R_{rs}(\lambda)$ variability (assessed over 20 min measurement cycles) between 3C and DD as a function of atmospheric optical state using data from three ports in the Western Channel. The greatest divergence between the algorithms occurs in the blue part of the spectrum where DD has significantly lower $R_{rs}(\lambda)$ variability than 3C in clearer sky conditions. We also consider $R_{rs}(\lambda)$ processing from a hypothetical two-sensor configuration (using only the $L_t(\lambda)$ spectroradiometer and the HSP and referred to as DD2) as a potential lower-cost measurement solution, which is shown to have comparable $R_{rs}(\lambda)$ and $R_{rs}(\lambda)$ variability to DD in clearer sky conditions. Our results support that the HSP sensor can fulfil a dual role in aquatic ecosystem monitoring by improving precision in $R_{rs}(\lambda)$ alongside its primary function to characterize aerosols.

**Keywords:** above-water reflectance; hyperspectral; solar irradiance; diffuse irradiance; autonomous monitoring; spectroradiometer; pyranometer

## 1. Introduction

The hyperspectral remote-sensing reflectance ($R_{rs}(\lambda)$ in units sr$^{-1}$ where $\lambda$ is the wavelength) of a water body contains information on a range of optically active water constituents such as chlorophyll-*a*, suspended matter and coloured dissolved organic matter (CDOM). Consequentially, time-series of $R_{rs}(\lambda)$ are a valuable component in aquatic ecosystem monitoring and interpretation of biogeochemical cycles. Moreover, In situ observations of $R_{rs}(\lambda)$ are vital to calibration and validation of water colour and related data products derived from satellite radiometers [1,2], which can be used to observe these properties at the global scale. Hyperspectral sensing of in situ $R_{rs}(\lambda)$ over the visible and near infra-red (NIR) allows calibration and validation of products derived from a wide range of satellite sensors with varying spectral response functions [3].

$R_{rs}(\lambda)$ is defined as the ratio of water-leaving radiance ($L_w(\lambda)$) to downwelling planar irradiance ($E_d(\lambda)$) just above the water surface. In situ $R_{rs}(\lambda)$ can be obtained in water and above water [4], with the latter most suitable for field surveys from ships and for

prolonged deployments to avoid rapid biofouling. With the exception of skylight-blocking procedures [5], $L_w(\lambda)$ cannot be measured directly above water and reflected skylight from the water surface has to be radiometrically corrected for when estimating $R_{rs}(\lambda)$. A central challenge in autonomous above-water monitoring of $R_{rs}(\lambda)$ is correcting for reflected skylight in non-ideal measurement conditions, particularly in the absence of independent information on water column and atmosphere optical properties [1,6–9]. Thus, for operational use of above-water systems, a high degree of autonomy is required in the downstream processing of $R_{rs}(\lambda)$; i.e., filtering anomalies or correcting for sub-optimal measurement conditions. Designing the system to minimize the capture of sun glint, either by selecting the most favourable viewing geometry from multiple radiance channels pointed in different azimuthal directions, or by mounting the radiance sensors on an axle rotated to compensate for solar and sensing platform azimuth, ensures that geometric constraints can be met over the course of a measurement day [7].

Above-water $R_{rs}(\lambda)$ (which we now refer to exclusively) is conventionally obtained using simultaneous, or in rapid sequence, measurements of $E_d(\lambda)$, sky radiance ($L_s(\lambda)$) typically at $40°$ viewing zenith, and total upwelling radiance ($L_t(\lambda)$) in the corresponding specular direction. For a perfectly flat water surface, it is valid to assume ideal specular reflection and $R_{rs}(\lambda)$ can be estimated from

$$R_{rs}(\lambda) \equiv \frac{L_w(\lambda)}{E_d(\lambda)} = \frac{L_t(\lambda)}{E_d(\lambda)} - \rho_f \frac{L_s(\lambda)}{E_d(\lambda)}, \tag{1}$$

where $\rho_f$ is the Fresnel reflectance factor ($\approx 0.0256$ at viewing zenith $40°$ for freshwater) and $\rho_f L_s(\lambda)/E_d(\lambda)$ represents specularly reflected skylight. In practice, $R_{rs}(\lambda)$ is typically estimated from an equation of the form

$$R_{rs}(\lambda) \equiv \frac{L_w(\lambda)}{E_d(\lambda)} = \frac{L_t(\lambda)}{E_d(\lambda)} - \rho_s \frac{L_s(\lambda)}{E_d(\lambda)} - \delta, \tag{2}$$

where $\rho_s$ is the effective Fresnel reflectance factor and $\delta$ (units $sr^{-1}$) is a scalar offset parameter [6,7,10,11]. Wind-roughening increases $\rho_s$ from $\rho_f$ [6,12,13] and forward-model approximations define the air–water surface slope distribution [14] as a function of wind speed [12]. Alternatively, $\rho_s$ can be solved for via various numerical optimization approaches [7,10,15]. For example, the "fingerprint" method empirically minimizes the propagation of atmospheric absorption features to estimate $\rho_s$ [7]. Then, $\delta$ accounts for residual sun glint and potential white cap contributions in rougher conditions [10]. This offset can be estimated dependent on the water type [16]; the near-infrared (NIR) $R_{rs}(\lambda)$ can be assumed to approach zero in clear ocean waters [17], or the "similarity spectrum" approach can be adopted based on constancy of $R_{rs}(\lambda)$ ratios in the NIR associated with pure water absorption bands [12].

The 3C (three-component glint model) method is a relatively recent development in $R_{rs}(\lambda)$ processing [11,15,18,19] which estimates $R_{rs}(\lambda)$ via spectral optimization of modelled atmospheric [20] and water [21] properties with respect to measured radiometric quantities [11,15,19]. Using 3C, the $R_{rs}(\lambda)$ equation is of the form

$$R_{rs}(\lambda) \equiv \frac{L_w(\lambda)}{E_d(\lambda)} = \frac{L_t(\lambda)}{E_d(\lambda)} - \rho_s \frac{L_s(\lambda)}{E_d(\lambda)} - \Delta(\lambda), \tag{3}$$

where

$$\Delta(\lambda) = \frac{\rho_{dd}}{\pi} \cdot \left( \frac{E_{dd}(\lambda)}{E_d(\lambda)} \right)^m + \frac{\rho_{ds}}{\pi} \cdot \left( \frac{E_{ds}(\lambda)}{E_d(\lambda)} \right)^m, \tag{4}$$

is a spectrally dependent offset parameter, $E_{dd}(\lambda)$ and $E_{ds}(\lambda)$ are the direct and diffuse irradiance components, $\rho_{dd}$ and $\rho_{ds}$ are direct and diffuse air–water reflectance factors, and the superscript $m$ notates that the irradiance ratios are model-optimized quantities. The factors of $1/\pi$ in Equation (4) ensure that each term has units $sr^{-1}$. In 3C, the spectral

ratios $L_s(\lambda)/E_d(\lambda)$, $(E_{dd}(\lambda)/E_d(\lambda))^m$ and $(E_{ds}(\lambda)/E_d(\lambda))^m$ comprise a set of three basis functions to correct the sky reflection and are combined as a linear sum weighted by optimized values of $\rho_s$, $\rho_{dd}$, and $\rho_{ds}$. The rationale is that each irradiance component ratio has a distinct spectral signature [22] which enables glint to be corrected for even when it has a different spectral shape to $L_s(\lambda)/E_d(\lambda)$. This occurs when there is angular asymmetry to the radiance distribution and the water surface is wind-roughened [5,11].

When it was introduced, the 3C method was proposed to be suitable for obtaining $R_{rs}(\lambda)$ in measurement conditions that can relax some geometric measurement constraints (e.g., sun-glint-affected data) as well as sub-optimal illumination and wind conditions [15]. This versatility has since been illustrated by exploring $R_{rs}(\lambda)$ dependencies in terms of varying wind speed [11], azimuthal viewing angles [11], light levels [19], and glint perturbations [19], whilst imposing only modest assumptions on the bio-optical properties of the water column. An inherent limitation of 3C is that the irradiance ratios are based on an atmospheric model that assumes a cloudless maritime atmosphere [20]. Consequently, optimization of the spectral shape of $(E_{dd}(\lambda)/E_d(\lambda))^m$ and $(E_{ds}(\lambda)/E_d(\lambda))^m$ (via the Angström exponent and atmospheric turbidity coefficient) can be non-physical in cloudier conditions.

In this study, we incorporate a novel hyperspectral pyranometer (HSP) which measures partitioned irradiance ($E_{dd}(\lambda)$ and $E_{ds}(\lambda)$) in a new variant of 3C-like $R_{rs}(\lambda)$ processing which, to distinguish from 3C, we refer to as the DD (direct-diffuse) method. The HSP irradiance spectra are combined with $L_s(\lambda)$ and $L_t(\lambda)$ spectra measured with a recently developed solar tracking radiometry platform (So-Rad) [23]. The HSP was originally designed with aerosol optical depth estimation in mind [24,25]. The device incorporates a shading pattern over multiple diffuser optics, allowing it to be operated without moving parts, compared to robotic alternatives, which makes it more suitable to be mounted on moving platforms. Here, it is used for two new purposes. First, it enables us to replace modelled irradiance ratios with measured ratios based on the true atmospheric optical state in Equation (4). Second, it enables us to characterize the atmospheric optical state and evaluate related $R_{rs}(\lambda)$ dependencies of 3C and DD.

Using data collected during deployment on a "ship-of-opportunity" over several months, we characterize the atmospheric dependence of differences between 3C and DD $R_{rs}(\lambda)$ and how these relate to optimized glint corrections. Due to the nature of the deployment, $R_{rs}(\lambda)$ reference measurements of on-water or in-water reflectance spectroscopy were not available alongside the above-water radiometry collected for this study. Consequently, our aims do not include a strict assessment of $R_{rs}(\lambda)$ retrieval accuracy, as has been the focus of previous work (e.g., [15]). Instead, a primary focus is analysis of variability in $R_{rs}(\lambda)$, compared between the two approaches, over multiple 20 min measurement cycles. Additionally, we benchmark against a (hypothetical) two-sensor version of DD (referred to as DD2) where the $L_s(\lambda)$ sensor is removed and the HSP is sole sky sensor, as a potentially cost-efficient alternative to the established three-sensor method for $R_{rs}(\lambda)$. Our results support that the HSP is a desirable addition within above-water $R_{rs}(\lambda)$ processing as it can improve precision in clearer conditions, whilst characterizing the atmospheric conditions known to greatly influence the satellite-based retrieval of water-leaving reflectance.

## 2. Field Data and Sensors

### 2.1. Field Data

Field data were collected onboard the Brittany Ferries *Armorique*, a vessel of 29,500 gross tonnage and 168 m in length. The dataset consists of 143 measurement days from three ports in the Western English Channel between 2 April–23 August 2020. The measurements were collected when the ship was stationary in the ports and near-shore at Cherbourg (83 days), Roscoff (55 days), and Plymouth (5 days). *Armorique* was moored in Cherbourg until 26 June 2020; thereafter, the majority of the data are from Roscoff.

The data analysis considers timestamps where the solar zenith angle was less than 60°, which is a typical threshold applied in satellite validation of $R_{rs}(\lambda)$ [26]. In early April this corresponds to an approximately 7 h time series, and in mid-June this corresponds to

an approximately 9.5 h time series. The data analysis then uses the subset of data when the ship was stationary as this reduces variability in $R_{rs}(\lambda)$ due to changing water properties. This includes all of the Cherbourg data where the ship was moored during movement restrictions in the 2020 pandemic. Median $R_{rs}(\lambda)$ for the three ports are shown in Figure 1, and all have a spectral maximum in the green part of the spectrum.

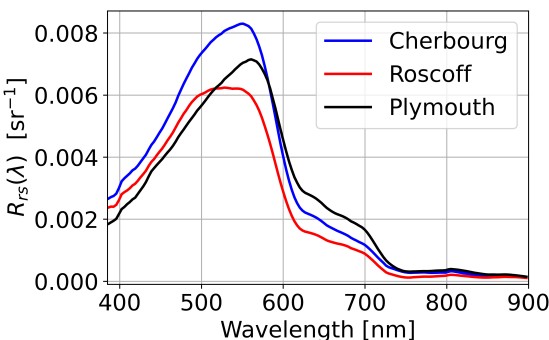

**Figure 1.** Median $R_{rs}(\lambda)$ from the three Western Channel ports computed using the 3C method (Section 3).

In all locations, the sensors had a view of water in the port and were kept in operation specifically for the purposes of this study. There was no regular maintenance of the sensors or their optical surfaces during this period. Upon retrieval of the sensors in August 2020, the sensors were found to be clean and in good condition, as confirmed by laboratory sensor calibrations.

### 2.2. The Solar-Tracking Radiometry Platform (So-Rad)

Radiance and irradiance spectra used to estimate $R_{rs}(\lambda)$ were measured using the autonomous Solar-tracking Radiometry platform (So-Rad) developed at Plymouth Marine Laboratory (Figure 2a) [23]. The measurements of $E_d(\lambda)$, $L_s(\lambda)$, and $L_t(\lambda)$ were made using TriOS RAMSES spectroradiometers triggered synchronously by the So-Rad. In the $R_{rs}(\lambda)$ processing in this study, $E_d(\lambda)$ measured by the HSP is used as a substitute for $E_d(\lambda)$ measured by the So-Rad. The TriOS sensors had a calibrated spectral range 320–950 nm supplied at a spectral resolution ~10 nm and sample spacing of 3.3 nm. $L_t(\lambda)$ and $L_s(\lambda)$ are a function of zenith viewing angle ($\theta_v$), azimuth viewing angle relative to the solar azimuth ($\Delta\phi$), and solar zenith angle ($\theta_s$). For particular measurement geometries, $L_t(\lambda)$ can be heavily contaminated by sun glint and, in ideal operation, So-Rad will avoid this by moving as near to the ideal $\Delta\phi = 135°$ [6] as possible within parent platform limitations. The radiance sensors are further mounted at $\theta_v = 40°$ to avoid platform shading and reflections. Schematic diagrams of the So-Rad measurement geometry are provided in [7], and can be cross-referenced with Figure 2a.

The So-Rad is designed for deployment on ships and large buoys, with precise registration of platform positioning and heading to maintain favourable viewing geometry. Taller mounting and large vessels are desirable to reduce sensor contamination and because the default configuration does not compensate for platform tilt. The So-Rad is a redesign of a previous autonomous reflectance platform [7] and allows low-cost and prolonged standalone operation (except cleaning) without need to calibrate viewing angles and allowing deployment using a modest (solar) power supply (15 W). The So-Rad provides open-source software support based on Python 3 running on a Linux operating system on a Raspberry Pi-3B [23].

In the specific deployment on *Armorique*, the $L_t(\lambda)$ and $L_s(\lambda)$ sensors were mounted on top of the bridge, starboard side, and were able to rotate from approximately 20° to 160° with respect to the bow–stern axis of the vessel (Figure 2c). As an example, when the vessel was stationary in Cherbourg, the ship heading (relative to north) was 19°, and $\Delta\phi$ was approximately between 40° and 180°. For the purposes of this study, the So-Rad was

configured to record regardless of the sun-avoidance possibilities provided within these angular bounds. In this experimental deployment, $\Delta\phi$ was not directly recorded due to a software configuration issue, although the daily time window influenced by glint can be inferred from the observation data. The $R_{rs}(\lambda)$ algorithms are thus tested under both optimized and sub-optimal conditions, depending on solar geometry, with quality-control filtering (Section 3.5) used to remove the most heavily glint contaminated spectra as well as poor illumination conditions or episodic disturbances such as precipitation.

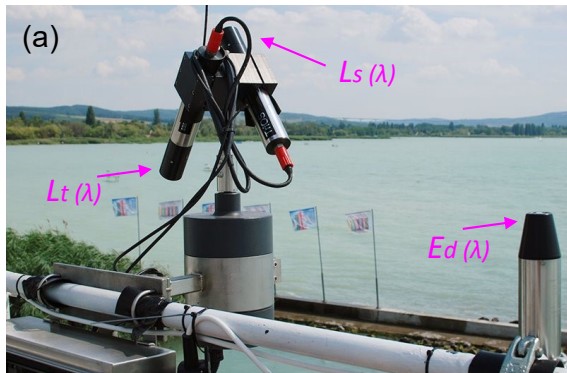 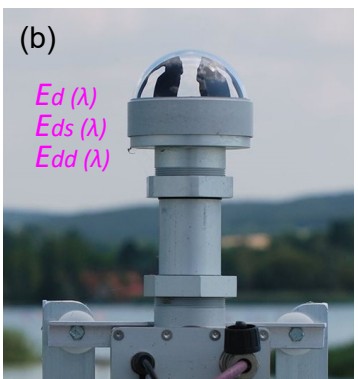

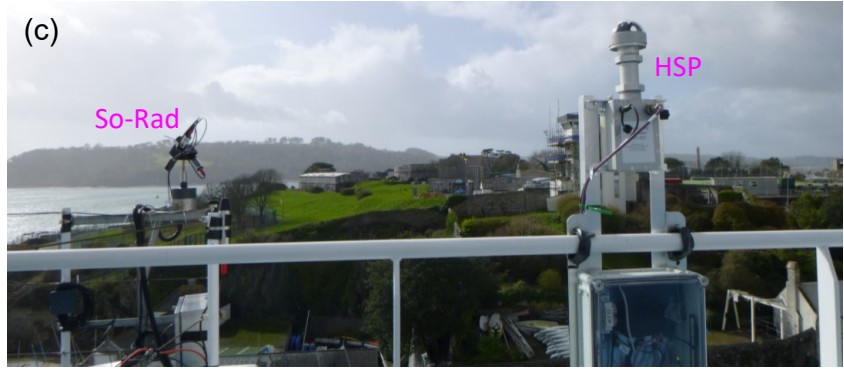

**Figure 2.** (**a**) Photograph of the Solar-tracking Radiometry sensor platform (So-Rad) used in $R_{rs}(\lambda)$ processing. The radiance ($L_s(\lambda)$ and $L_t(\lambda)$) and irradiance ($E_d(\lambda)$) sensors are indicated with the radiance sensors oriented at $\pm40°$ zenith and mounted to a rotating platform. (**b**) Photograph of the prototype hyperspectral pyranometer (HSP) sensor deployed alongside So-Rad. The HSP measures partitioned irradiance ($E_{dd}(\lambda)$ and $E_{ds}(\lambda)$) and in this study is used as a substitute $E_d(\lambda)$ sensor in $R_{rs}(\lambda)$ processing. (**c**) Photograph of the sensor positioning during the *Armorique* deployment.

### 2.3. The Hyperspectral Pyranometer (HSP)

The hyperspectral pyranometer (HSP) provides both an alternative measurement of $E_d(\lambda)$ to the TriOS irradiance sensor and the measuring of the diffuse component of downwelling irradiance, $E_{ds}(\lambda)$. The direct irradiance component can then be determined from $E_{dd}(\lambda) = E_d(\lambda) - E_{ds}(\lambda)$. The HSP therefore measures direct-diffuse "partitioned" downwelling irradiance. A photograph of the HSP is shown in Figure 2b. The HSP incorporates multiple diffuser channels operating simultaneously under varying degrees of shade cast onto the sensor surface using an integrated shading pattern. Of particular significance to operation on moving vessels is that the arrangement of the fore-optics and shading poses no limitations on the azimuthal orientation of the sensor, and no moving parts are needed. Wood et al. [24] outline a predecessor to the HSP used in this study and describe its application to measuring direct-diffuse solar irradiance in a marine environment.

On *Armorique*, the HSP and So-Rad sensors were separated horizontally by approximately 10 m (Figure 2b). The HSP was secured in an elevated position to ensure that its

field of view was unobstructed. In this study, the TriOS $E_d(\lambda)$ is replaced by the HSP $E_d(\lambda)$ in the $R_{rs}(\lambda)$ processing to mimic the three-sensor configuration of the So-Rad. The spectral measurement range of the HSP used in this study is 350–1150 nm, which was spectrally calibrated over approximately 380–950 nm and had a spectral resolution of $\approx$3.3 nm (improved from Wood et al. [24]). The HSP calibration was checked against an FEL standard calibration lamp (UK National Physical Laboratory) shortly after the end of the deployment. This showed that the calibration was within 5% of the reference lamp over the range used in this study, with the largest calibration difference at blue wavelengths.

The HSP was originally developed with estimation of aerosol optical depth (AOD) as a primary objective [24,25]. In clearer skies, spectrally resolved AOD can be estimated either from the atmospheric attenuation of $E_{dd}(\lambda)$ (direct beam method [24]) or from the ratio $E_{ds}(\lambda)/E_d(\lambda)$ (diffuse ratio method [25]). The presence of clouds also impacts on $E_{ds}(\lambda)/E_d(\lambda)$ which tends toward unity in overcast conditions, and rapidly fluctuates in scattered-cloud conditions.

In this study, we characterize the atmospheric optical state using the spectrally integrated diffuse ratio (IDR) defined by

$$IDR = \frac{\int_{\lambda_i}^{\lambda_f} E_{ds}(\lambda)d\lambda}{\int_{\lambda_i}^{\lambda_f} E_d(\lambda)d\lambda},$$　　　　　　(5)

where $\lambda_i$ = 385 nm and $\lambda_f$ = 900 nm, with the wavelength limits based on filtering of the spectra (Section 2.4). The measured IDR for the Western Channel deployment varies between $\approx$0.04 and $\approx$0.99, with the lowest value representing the clearest skies and the highest value the most overcast. Illustrative examples of HSP measurements of $E_{ds}(\lambda)$ and $E_d(\lambda)$ and the IDR in different measurement conditions are shown in Figure 3. For readability, we refer to IDR < 0.2 as "clear conditions", 0.2 < IDR < 0.8 as "intermediate conditions", and IDR > 0.8 as "overcast conditions". Intermediate conditions include both scattered cloud (associated with higher variability in the IDR) and haze (associated with lower variability in the IDR).

The spectral shape of $E_d(\lambda)$ is generally similar across all atmospheric conditions [7]. To understand the relative magnitudes of $E_{ds}(\lambda)$ and $E_d(\lambda)$ in Figure 3, the following physical principles apply. For the clear-sky conditions, $E_{ds}(\lambda)$ is entirely derived from Rayleigh and aerosol scattering. As cloud starts to build across the sky, it is a highly scattering medium with minimal absorption. This means it adds more scattered radiation on top of the Rayleigh and aerosol, increasing the $E_{ds}(\lambda)$ at the expense of $E_{dd}(\lambda)$. Only when cloud cover is high and the clouds are very thick does cloud absorption start to dominate. In turn, this reduces $E_{ds}(\lambda)$ (and hence $E_d(\lambda)$ as $E_{dd}(\lambda)$ tends to zero in these conditions) back down to comparable or lower irradiances than the clear-sky case. The examples in Figure 3 are also from different months during the deployment, with different solar zenith angles influencing the relative irradiance magnitudes.

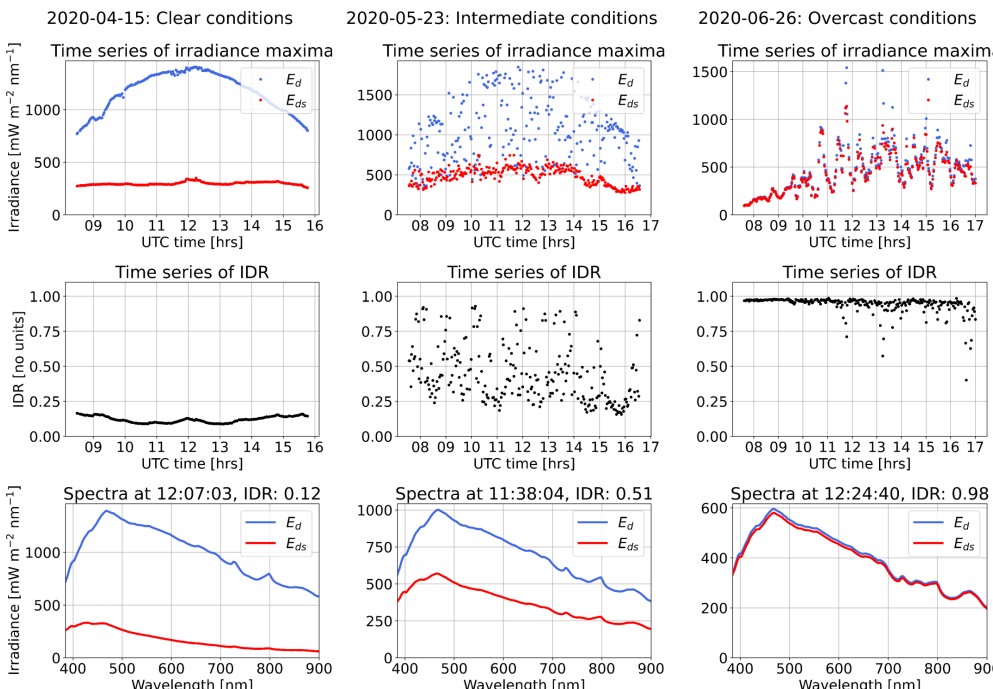

**Figure 3.** Examples of atmospheric optical state characterization using HSP measurements of $E_{ds}(\lambda)$, $E_d(\lambda)$ and the IDR, Equation (5). Left column: clear conditions; centre column: intermediate (predominately scattered-cloud) conditions; right column: overcast conditions. The $E_{ds}(\lambda)$ and $E_d(\lambda)$ spectra are filtered using 70 nm Savitzky–Golay filter (Section 2.4). The time series are for solar zenith angle $< 60°$. The date format is YYYY-MM-DD.

### 2.4. Data Combination Prior to $R_{rs}(\lambda)$ Processing

Co-registered HSP (sample rate 1 min) and So-Rad (sample rate up to 30 s) (ir)radiance measurements for $R_{rs}(\lambda)$ processing were defined as those with timestamps at the midpoint of integration intervals matching to within 30 s. The timestamp matching resulted in a mean sensor time difference between HSP and So-Rad observations of approximately 12 s across the dataset.

Spectral filtering was applied prior to $R_{rs}(\lambda)$ processing to better match the resolution and spectral response between sensor types. This is necessary as Equation (3) incorporates spectral ratios of the two sensor measurements. To perform this, the HSP spectra (sample spacing 1 nm) were downsampled to the wavelength spacing adopted for the TriOS sensors (3.3 nm). The set of spectra from both sensors were then smoothed using a third-order Savitzky–Golay filter, a class of convolutional filter that smooths the spectra without distorting the signal tendency. Following Simis and Olsson [7], a 70 nm filter width was used, as smaller filter widths were less able to adequately suppress high-frequency spikes in spectral ratios. The combined HSP-TriOS spectra are defined on [385, 900] nm, with the wavelength range sufficient to remove edge effects from the spectral filter.

## 3. Reflectance Processing

### 3.1. Overview of $R_{rs}(\lambda)$ Algorithm Variants

This study considers three $R_{rs}(\lambda)$ algorithms that are variants of the existing 3C spectral optimization:

1.　3C (three-component glint). Conventional 3C processing following the implementation in Groetsch et al. [15] where a water model [21] and an atmospheric model [20] are both used in the spectral optimization. The observed spectral ratios are $L_s(\lambda)/E_d(\lambda)$ and $L_t(\lambda)/E_d(\lambda)$, and the model-optimized ratios are $(E_{ds}(\lambda)/E_d(\lambda))^m$ and $(E_{dd}(\lambda)/E_d(\lambda))^m$.

2.  DD (direct-diffuse). This is 3C-like processing using the water model but not the atmospheric model, and extending the observed spectral ratios to be $L_s(\lambda)/E_d(\lambda)$, $L_t(\lambda)/E_d(\lambda)$, $E_{ds}(\lambda)/E_d(\lambda)$, and $E_{dd}(\lambda)/E_d(\lambda)$.
3.  DD2 (direct-diffuse with two sensors). This is 3C-like processing using the water but not the atmospheric model and using $L_t(\lambda)/E_d(\lambda)$, $E_{ds}(\lambda)/E_d(\lambda)$, and $E_{dd}(\lambda)/E_d(\lambda)$ as observed spectral ratios (i.e., removing the $L_s(\lambda)$ sensor). This corresponds to measurements from a hypothetical two-sensor system ($L_t(\lambda)$ spectroradiometer and HSP pyranometer for $E_d(\lambda)$, $E_{dd}(\lambda)$, and $E_{ds}(\lambda)$).

For the comparative study of the algorithm variants it is convenient to express Equations (3) and (4) in the form

$$R_{rs}(\lambda) = \frac{L_t(\lambda)}{E_d(\lambda)} - R_g(\lambda), \tag{6}$$

where

$$R_g(\lambda) = R_s(\lambda) + R_{ds}(\lambda) + R_{dd}(\lambda), \tag{7}$$

represents the total glint correction (total surface-reflected radiance), and

$$R_s(\lambda) = \begin{cases} \rho_s \frac{L_s(\lambda)}{E_d(\lambda)} & 3C, DD \\ 0, & DD2 \end{cases} \tag{8}$$

is the specular glint component associated with $L_s(\lambda)$, and

$$R_{dd}(\lambda) = \begin{cases} \frac{\rho_{dd}}{\pi} \cdot \left( \frac{E_{dd}(\lambda)}{E_d(\lambda)} \right)^m & 3C \\ \frac{\rho_{dd}}{\pi} \cdot \frac{E_{dd}(\lambda)}{E_d(\lambda)}, & DD, DD2 \end{cases} \tag{9}$$

is the direct glint (sun glint) component associated with $E_{dd}(\lambda)$, and

$$R_{ds}(\lambda) = \begin{cases} \frac{\rho_{ds}}{\pi} \cdot \left( \frac{E_{ds}(\lambda)}{E_d(\lambda)} \right)^m & 3C \\ \frac{\rho_{ds}}{\pi} \cdot \frac{E_{ds}(\lambda)}{E_d(\lambda)}, & DD, DD2 \end{cases} \tag{10}$$

is the diffuse glint (sky glint) component associated with $E_{ds}(\lambda)$. We remind the reader that $m$ indicates a model-optimized quantity. $R_g(\lambda)$, $R_s(\lambda)$, $R_{dd}(\lambda)$, and $R_{ds}(\lambda)$ have units $\text{sr}^{-1}$ and are used to compare the glint corrections between the three algorithms.

### 3.2. Water and Atmospheric Models Used in Spectral Optimization

Both the water and atmospheric models in the spectral optimization have previously been outlined by Groetsch et al. [15]. These are unchanged in this study and here we give an overview. The spectral optimization is designed to enable the separation of water and atmospheric signal components, rather than to perform a water parameter inversion [15].

The water model, based on Albert and Mobley [21], is used to model remote-sensing reflectance (notated as $R_{rs}^m(\lambda)$ to distinguish from $R_{rs}(\lambda)$ estimated in the optimization) based on a set of inherent optical properties. The computation of $R_{rs}^m(\lambda)$ from water absorption and optical backscatter is summarized by Equations (7)–(12) by Groetsch et al. [15], and assumes that the water is optically complex (Case 2) water (i.e., the water constituent concentrations may vary independently). The water-column absorption has dependence on the absorption spectrum of pure water [27], the specific absorption of phytoplankton [28], the concentration of chlorophyll-*a* ($C_{chl}$), CDOM absorption slope referenced to 440 nm ($S_y$), and CDOM absorption referenced to 440 nm ($C_y$). The optical backscatter incorporates pure water and suspended matter components and has dependence on the concentration of suspended matter ($C_{sm}$).

In principle, the water model within the optimization can be modified for local water conditions (e.g., by using a different spectrum for specific absorption of phytoplankton,

or considering detritus/minerals absorption) [15,19]. Here, we adopt the default water model in 3C, as we are interested in benchmarking what has been used in previous applications [11,15,18,19]. It is noted that that the water model we use does not account for azimuth dependence, which results in bidirectional variation of $R_{rs}(\lambda)$ [29].

The atmospheric model, based on Gregg and Carder [20], is used to model the irradiance ratios $(E_{dd}(\lambda))/E_d(\lambda))^m$ and $(E_{ds}(\lambda)/E_d(\lambda))^m$ and assumes a cloudless maritime atmosphere. It is noted that only the irradiance ratios are defined with the available data (not the absolute values). The numerator of $(E_{ds}(\lambda)/(E_d(\lambda))^m$ is the modelled sum of aerosol- and Rayleigh-scattered irradiance, whereas $E_{ds}(\lambda)$ measured by the HSP corresponds to measurement of both diffuse sub-components at once. There are two free parameters in the spectral optimization of the irradiance ratios: $\alpha$ (Angström exponent) and $\beta$ (turbidity coefficient) which relate to AOD, $\tau_a$, via the power law relationship $\tau_a(\lambda) = \beta(\lambda/\lambda_a)^{-\alpha}$, where $\lambda_a$ = 550 nm is a reference wavelength. $\alpha$ and $\beta$ impact on the spectral shape of $(E_{dd}(\lambda))/E_d(\lambda))^m$ and $(E_{ds}(\lambda)/E_d(\lambda))^m$ and the governing equations are given by (14)–(19) in Groetsch et al. [15].

The varying parameters in the spectral optimization are summarized in Table 1. The previous implementation by Groetsch et al. [15] experimented with using a constant and varying $S_y$ in the optimization, showing that both could be justified. As $S_y$ is not known for our dataset, we leave this a varying (but bounded) parameter.

**Table 1.** Initial conditions (IC) and parameter bounds for varying parameters in spectral optimization based on previous values in Groetsch et al. [15]. Further fixed parameters (e.g., additional conditions within the atmospheric model) are also set to the defaults in Groetsch et al. [15].

| Parameter Group | Parameter Name | Symbol and Units | IC [min, max] |
|---|---|---|---|
| Water properties | Chlorophyll-a concentration | $C_{chl}$ [mg m$^{-3}$] | 5 [0.01, 100] |
| | CDOM absorption at 440 nm | $C_y$ [nm$^{-1}$] | 0.1 [0.01, 5] |
| | CDOM absorption slope | $S_y$ [nm$^{-1}$] | 0.012 [0.01, 0.02] |
| | Concentration of SPM | $C_{sm}$ [g m$^{-3}$] | 10 [0.0, 100] |
| Atmospheric properties | Ångström exponent | $\alpha$ [-] | 1 [0, 3] |
| | Turbidity | $\beta$ [-] | 0.05 [0, 10] |
| Interfacial reflectance | Air-water reflectance factor | $\rho_s$ [-] | $\rho_f$ [0, 0.1] |
| | Direct air–water reflectance factor | $\rho_{dd}$ [-] | 0 [0, 0.1] |
| | Diffuse air–water reflectance factor | $\rho_{ds}$ [-] | 0 [0.01, 0.1] |

### 3.3. Spectral Optimization Procedure

The spectral optimization in 3C, DD, and DD2 is formulated in terms of a spectral minimization of the residual sum of squares between observed and modelled $L_t(\lambda)/E_d(\lambda)$ of the form

$$\epsilon = \sum_i W(\lambda_i)\left[\frac{L_t(\lambda_i)}{E_d(\lambda_i)} - \left(\frac{L_t(\lambda_i)}{E_d(\lambda_i)}\right)^m\right]^2, \tag{11}$$

where $\epsilon$ is a residual parameter, $W(\lambda_i)$ is a spectral weights vector, and $i$ is a spectral index which runs over all wavelength bins. $(L_t(\lambda)/E_d(\lambda))^m$ is derived by rearranging Equation (7) to give

$$\left(\frac{L_t(\lambda)}{E_d(\lambda)}\right)^m = R_{rs}^m(\lambda) + R_s(\lambda) + R_{ds}(\lambda) + R_{dd}(\lambda), \tag{12}$$

where $R_s(\lambda)$, $R_{ds}(\lambda)$, $R_{dd}(\lambda)$ are given by Equations (8)–(10) and we remind the reader that $R_{rs}^m(\lambda)$ is model-optimized. The spectral minimization of Equation (11), subject to the optimization of parameters in Table 1, uses the limited memory Broyden–Fletcher–Goldfarb–Shano algorithm as the numerical solver [30]. Once the optimization has been performed, $\rho_s$, $\rho_{dd}$, and $\rho_{ds}$ are substituted in Equations (8) and (9) to derive optimized glint components $R_s(\lambda)$, $R_{ds}(\lambda)$, $R_{dd}(\lambda)$ that are then substituted in Equation (7) to derive optimized $R_{rs}(\lambda)$.

The residual parameter $\epsilon$ has been used to flag anomalous $R_{rs}(\lambda)$ retrievals [11,15,19], and has been interpreted as a proxy for $R_{rs}(\lambda)$ retrieval accuracy [19]. $W(\lambda_i)$ is discussed further in Groetsch et al. [15]. Following a default configuration, we set $W(\lambda_i) = 1$, except over 370–500 nm where $W(\lambda_i) = 2$ and over 760–770 nm where $W(\lambda_i) = 0.1$, the latter mitigating sensitivity to the deep oxygen-A rotational absorption line. The higher weight in the blue part of the spectrum focuses the optimization on heavily sky-light-affected spectral regions with little biological influence and reduces dependence on the bio-optical model parameterization [15].

### 3.4. Examples of Reflectance Processing and Glint Corrections

We now give illustrative examples of the reflectance processing and glint corrections for 3C, DD, and DD2 in clear (Figure 4), intermediate (Figure 5), and overcast (Figure 6) conditions. Each example compares measured $L_t(\lambda)/E_d(\lambda)$ and model-optimized $(L_t(\lambda)/E_d(\lambda))^m$ which define the residual in Equation (11), alongside derived $R_{rs}(\lambda)$. The examples also illustrate that the algorithms can use very different combinations of the optimized glint components $R_s(\lambda)$, $R_{dd}(\lambda)$, and $R_{ds}(\lambda)$ that are used to form $R_g(\lambda)$ for each algorithm (we remind the reader that $R_s(\lambda) = 0$ in DD2). The measured $(L_s(\lambda)/E_d(\lambda)$, $E_{dd}(\lambda)/E_d(\lambda)$, $E_{ds}(\lambda)/E_d(\lambda))$ and model-optimized $((E_{dd}(\lambda)/E_d(\lambda))^m$, $(E_{ds}(\lambda)/E_d(\lambda))^m)$ spectral ratios that determine the spectral shape of $R_s(\lambda)$, $R_{dd}(\lambda)$, and $R_{ds}(\lambda)$, Equations (8)–(10), are also shown and are sometimes referred to as "glint basis functions". It is important to note that Figures 4–6 are just a few examples. A statistical range of outcomes are present for algorithm differences in $R_{rs}(\lambda)$ and the optimized glint components $R_s(\lambda)$, $R_{dd}(\lambda)$, and $R_{ds}(\lambda)$, which we characterize as a function of IDR in the results.

In the examples from clear and intermediate conditions (Figures 4 and 5), the measured spectral ratios $(L_s(\lambda)/E_d(\lambda)$, $E_{ds}(\lambda)/E_d(\lambda)$, and $E_{ds}(\lambda)/E_d(\lambda))$ have spectral curvature, whereas in overcast skies (Figure 6) the measured spectral ratios tend toward being spectrally flat. The examples illustrate that the algorithms often use very different combinations of $R_s(\lambda)$, $R_{dd}(\lambda)$, and $R_{ds}(\lambda)$ to form a similar overall glint correction, $R_g(\lambda)$. For example, in Figure 4, 3C predominately uses $R_s(\lambda)$ and $R_{ds}(\lambda)$, whereas DD uses $R_{dd}(\lambda)$ and $R_{ds}(\lambda)$. The examples also serve to illustrate that 3C processing can result in $(E_{dd}(\lambda)/E_d(\lambda))^m$ and $(E_{ds}(\lambda)/E_d(\lambda))^m$ that have a similar (Figure 4) and dissimilar (Figure 6) shape to measured $E_{dd}(\lambda)/E_d(\lambda)$ and $E_{ds}(\lambda)/E_d(\lambda)$. In addition to the 3C atmospheric model [20] being defined for clear conditions, there are two further explanations why it is reasonable for the model-optimized curves to differ from measured, both previously raised by Groetsch et al. [11] and Groetsch et al. [15]. First, as is the case in Figures 5 and 6, $R_{dd}(\lambda)$ and $R_{ds}(\lambda)$, can be small compared to $R_s(\lambda)$. The spectral shape of model-optimized $R_{dd}(\lambda)$ and $R_{ds}(\lambda)$ is therefore often a second-order correction that is being model-optimized. Second, the sum of "non-physical" and $R_{dd}(\lambda)$ and $R_{ds}(\lambda)$ can combine to give a physically accurate resultant curve. For example, as illustrated in Figure 6, the sum of $(E_{dd}(\lambda)/E_d(\lambda))^m$ and $(E_{ds}(\lambda)/E_d(\lambda))^m$, which have opposite curvature, will give a flat resultant glint curve.

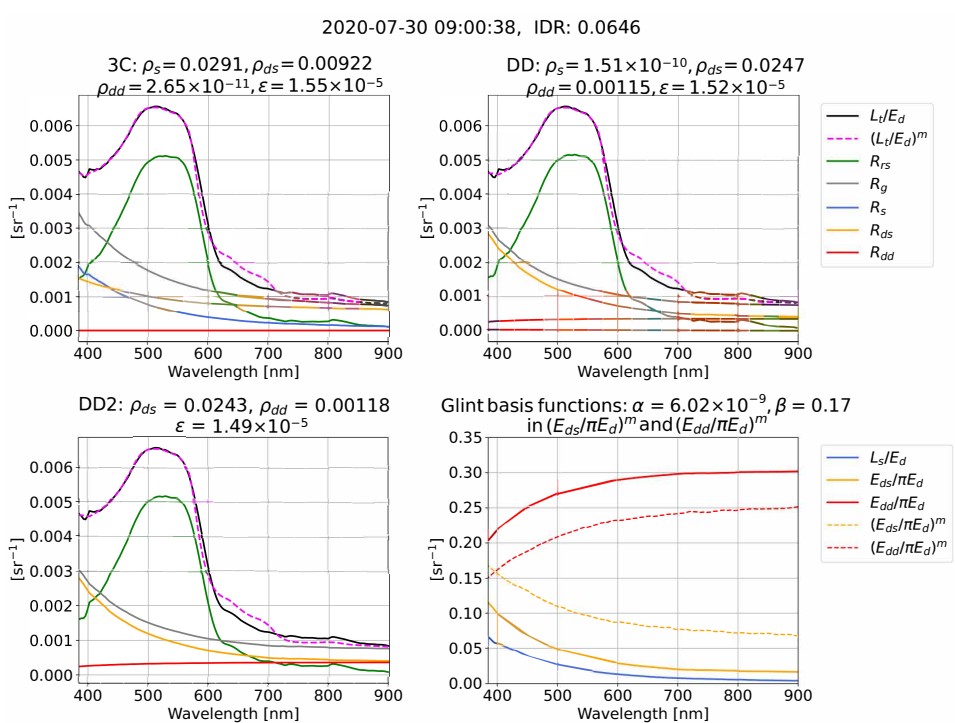

**Figure 4.** Example of $R_{rs}(\lambda)$ processing for 3C, DD, and DD2 in clear conditions from Roscoff. The date format is YYYY-MM-DD.

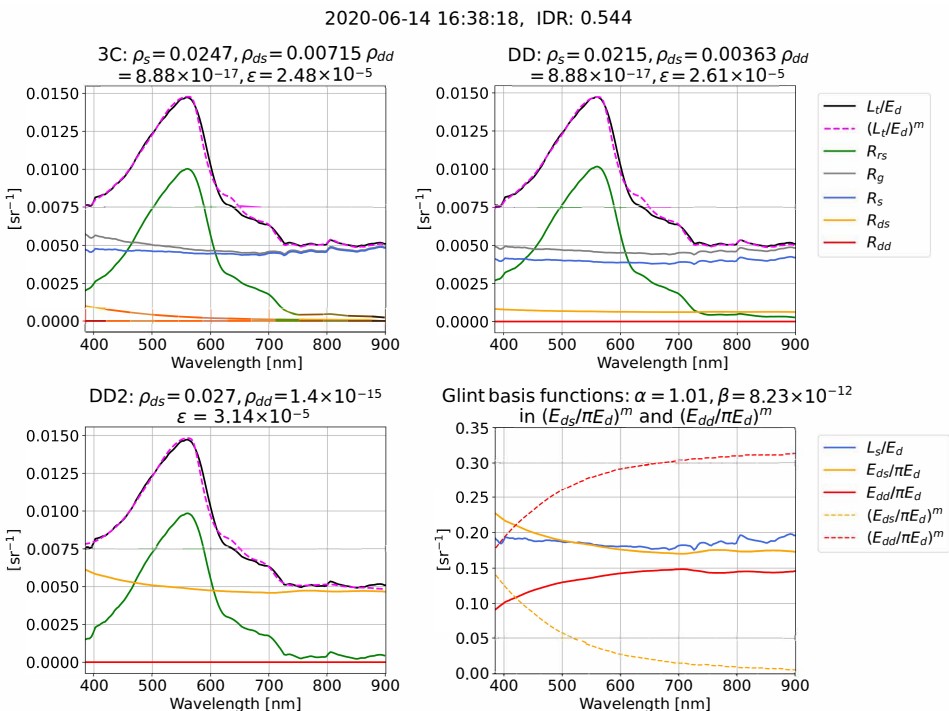

**Figure 5.** Example of $R_{rs}(\lambda)$ processing for 3C, DD, and DD2 in intermediate conditions from Cherbourg. The date format is YYYY-MM-DD.

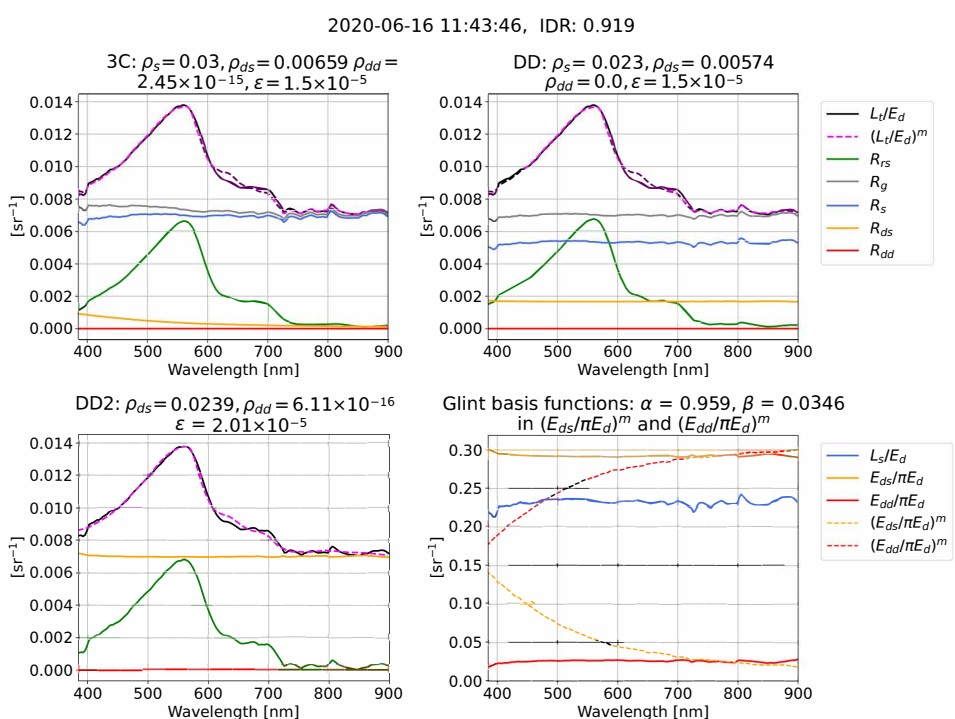

**Figure 6.** Example of $R_{rs}(\lambda)$ processing for 3C, DD, and DD2 in overcast conditions from Cherbourg. The date format is YYYY-MM-DD.

### 3.5. Quality Control

The $R_{rs}(\lambda)$ quality control (QC) chain is provided on the *monda* (MONocle Data Analysis) repository [31] and is implemented in three stages. First, a set of basic radiometric filters were applied to measured $E_d(\lambda)$, $L_s(\lambda)$, and $L_t(\lambda)$. This includes setting a minimum value on the spectral maximum of $E_d(\lambda)$ (500 mW m$^{-2}$nm$^{-1}$), and removing glint-affected spectra when $L_t(\lambda)/E_d(\lambda)$ exceeds an empirical threshold of 0.025 sr$^{-1}$ on the interval 850–900 nm [15]. Second, timestamps where the algorithm optimizations terminated on the upper bounds for $\rho_s$ (3C and DD), $\rho_{dd}$ and $\rho_{ds}$ (all algorithms), shown in Table 1, were removed as these resulted in non-physical $R_{rs}(\lambda)$. Third, timestamps were removed where the $R_{rs}(\lambda)$ spectral maximum was less than 0.005 sr $^{-1}$. The subset of timestamps where 3C, DD, and DD2 all pass QC were then used in the data analysis. The analysis is also restricted to solar zenith angles less than 60°. In previous applications of 3C, the distribution of the residual $\epsilon$, Equation (11), has been used to filter outliers. This was not performed here.

### 3.6. Computation of $R_{rs}(\lambda)$ Variability

A central aspect of this study is the assessment of $R_{rs}(\lambda)$ variability, which we measure using the coefficient of variation, $\text{CV}[R_{rs}(\lambda)]$ (ratio of standard deviation to mean). $\text{CV}[R_{rs}(\lambda)]$ was computed using a rolling window of length 20 min to first compute the rolling mean and rolling standard deviation (STD). $\text{CV}[R_{rs}(\lambda)]$ is typically used as a variability metric in aquatic remote sensing, as it enables comparison of relative variability in $R_{rs}(\lambda)$ between spectral bands. It is also often interpreted as a proxy for measurement precision (e.g., [11]), as it indicates dispersion (specifically, the random error component in $R_{rs}(\lambda)$ scaled by the mean, if it is assumed that the distribution of $R_{rs}(\lambda)$ is statistically stationary). In the analysis, we use the term "improved precision" to describe lower $\text{CV}[R_{rs}(\lambda)]$ when comparing algorithms, as it informs us of relative variation in $R_{rs}(\lambda)$ due to how the numerical optimizations differ. For each time window, the rolling mean and STD of the IDR were also computed, enabling $\text{CV}[R_{rs}(\lambda)]$ to be related to the atmospheric conditions.

Due to the 1 min sample spacing used in the analysis, there are 20 data points in a window when all of the timestamps pass QC (Section 3.5). However, due to QC failure, there can be data gaps, and a threshold of > 8 good quality data points (QC passes)

within a window was used in generating the statistics. The window length of 20 min represents a trade-off between being long enough to ensure that there are sufficient data points to estimate $CV[R_{rs}(\lambda)]$, and short enough to ensure that variability in $R_{rs}(\lambda)$ due to changing solar angle, tidal influence, or inherent optical water properties is small. Relative comparison of $CV[R_{rs}(\lambda)]$ between algorithms is a robust comparison to make as the additional components of variability listed above can be assumed to be the same in each case. Additionally, the window size of 20 min is significantly less than typical (∼1 h or greater) time window lengths used in satellite validation of $R_{rs}(\lambda)$ [32].

## 4. Results

### 4.1. Atmospheric Optical State

The integrated diffuse ratio (IDR), Equation (5), is used as an atmospheric optical state variable when evaluating $R_{rs}(\lambda)$, and is shown for the Western Channel deployment in Figure 7a. The data are divided into 10 discrete bins between 0 and 1 with bin width of 0.1 IDR. Atmospheric categories used in the data description follow Section 2.3. Figure 7a indicates a spread of the data across the potential range of atmospheric radiance distributions with most observations obtained under clear conditions (defined here as IDR < 0.2).

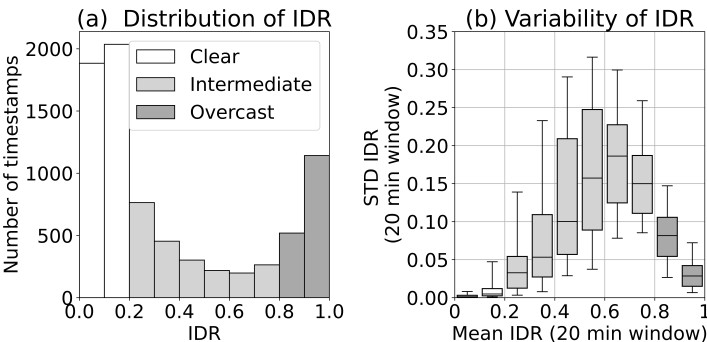

**Figure 7.** Atmospheric optical conditions for Western Channel deployment. (**a**) Distribution of integrated diffuse ratio (IDR) for all timestamps that have passed quality control (7782 timestamps in total). (**b**) Relationship between standard deviation (STD) of the IDR and mean IDR calculated using a 20 min rolling window (5840 windows in total). The upper and lower box limits notate upper and lower quartiles, the central line indicates the median values, and the whiskers correspond to the 5th and 95th percentiles.

The dependence of IDR variability on the IDR, computed following Section 3.6, is shown in Figure 7b. Clear conditions have the lowest variability in IDR, followed by overcast, with intermediate conditions having the highest variability. Scattered cloud conditions, as shown in Figure 3, are associated with the highest STD IDR. In the analysis of $R_{rs}(\lambda)$ we do not explicitly look at dependencies in terms of IDR variability. However, due to the defined relationships in Figure 7b, an IDR value range can be referenced to a range for IDR variability.

### 4.2. Atmospheric Dependence of $R_{rs}(\lambda)$ Variability

The $R_{rs}(\lambda)$ statistics are assessed at four wavelengths: 400 nm (blue), 560 nm (green), 665 nm (red), and 865 nm (NIR). These are informed by the spectral bands of multispectral satellite radiometers and are called "bands" in the analysis. The dependence of $CV[R_{rs}(\lambda)]$ on the IDR is shown for 3C, DD, and DD2 in Figure 8. In the data description we compare 3C and DD (assessment of difference between model-optimized or measured irradiance ratios in the glint correction), or between DD and DD2 (assessment of the effect of removing the sky radiance sensor).

In the 400 nm band, 3C has considerably higher $CV[R_{rs}(\lambda)]$ than DD in clearer conditions (median $CV[R_{rs}(\lambda)] \approx 50\%$ higher and upper quartiles $\approx 100\%$ higher). However,

in intermediate conditions, $CV[R_{rs}(\lambda)]$ can be slightly larger for DD than 3C. In the 560, 665, and 865 nm bands, 3C and DD have very similar $CV[R_{rs}(\lambda)]$ and overall dependence of $CV[R_{rs}(\lambda)]$ on the IDR. For these bands, $CV[R_{rs}(\lambda)]$ is lowest in clear conditions and highest in intermediate conditions. An explanation for this behaviour is that intermediate conditions typically correspond to an angularly inhomogeneous radiance distribution, and the measurement of $L_s(\lambda)$ will be generally less spectrally representative of surface-reflected radiance. On the other hand, in clearer and overcast conditions, which typically have a more angularly homogeneous radiance distribution, measurements of $L_s(\lambda)$ will be more representative of surface-reflected radiance.

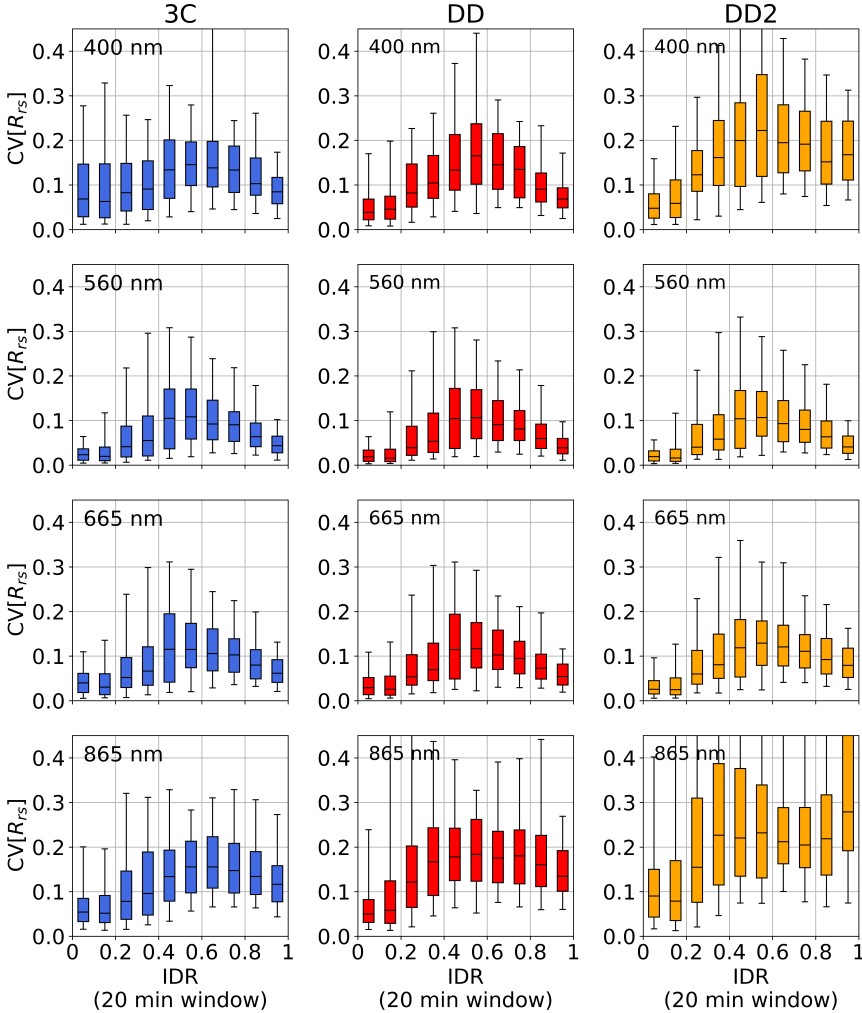

**Figure 8.** Atmospheric dependence of $R_{rs}(\lambda)$ variability ($CV[R_{rs}(\lambda)]$ computed using a 20 min rolling window). Rows: spectral bands; columns: algorithm variants. The upper and lower box limits notate upper and lower quartiles, the central line indicates the median values, and the whiskers correspond to the 5th and 95th percentiles. The most notable difference in $CV[R_{rs}(\lambda)]$ between 3C and DD is the 400 nm band (top row) in clear atmospheric conditions (IDR < 0.2), which are also the most common measurement conditions in the deployment (Figure 7a).

In the 560 and 665 nm bands, DD2 and DD have similar $CV[R_{rs}(\lambda)]$. However, in the 400 and 865 nm bands in intermediate and overcast conditions, DD2 has significantly greater $CV[R_{rs}(\lambda)]$ (ranging from $\approx$ 20–100% higher). $CV[R_{rs}(\lambda)]$ generally has a less defined peak in intermediate conditions for DD2 than DD and 3C, and is often equally high in overcast conditions (e.g., at 400 nm in Figure 8). This is thought to be due to DD2 becoming increasing reliant on $R_{ds}(\lambda)$ in overcast conditions, which results in a higher fit residual (see Section 4.4).

### 4.3. Atmospheric Dependence of $R_{rs}(\lambda)$ Differences

The assessment of $R_{rs}(\lambda)$ differences between algorithms is performed on a timestamp-by-timestamp basis for the IDR bins in Figure 7a. We remind the reader that this is not an assessment of retrieval accuracy as reference $R_{rs}(\lambda)$ is not available. Instead, the aim is to establish the atmospheric conditions where similar and different $R_{rs}(\lambda)$ retrievals occur and then relate to the optimized glint components (Section 4.4).

The atmospheric dependence of absolute and percentage differences for $R_{rs}^{DD}(\lambda) - R_{rs}^{3C}(\lambda)$ are shown in Figure 9. The 400, 560, and 665 nm bands all have the general trend that $R_{rs}^{DD}(\lambda) - R_{rs}^{3C}(\lambda)$ is greatest in overcast conditions. This is most notable for the 400 nm band, where the median $R_{rs}^{DD}(\lambda) - R_{rs}^{3C}(\lambda)$ is $\approx 10\%$ in the most overcast conditions. It is also noted that even in clear conditions, there is large spread of outlying $R_{rs}^{DD}(\lambda) - R_{rs}^{3C}(\lambda)$ (represented by the whiskers in Figure 9). It is anticipated that the statistical difference between $R_{rs}^{DD}(\lambda) - R_{rs}^{3C}(\lambda)$ increases with IDR, as the 3C model approximation is based on clear-sky conditions and becomes increasingly less accurate in overcast conditions. An explanation for why $R_{rs}^{DD}(\lambda) - R_{rs}^{3C}(\lambda) > 0$ in overcast conditions is due to the "false curvature" of model-optimized $(E_{ds}(\lambda)/E_d(\lambda))^m$ that can occur (e.g., see comparison with the measurements in Figure 6). Absolute and percentage differences are lowest in the 560 and 665 nm bands. Due to low absolutes values of $R_{rs}(\lambda)$, the 865 nm band has small absolute differences for $R_{rs}^{DD}(\lambda) - R_{rs}^{3C}(\lambda)$ but large percentage differences.

The atmospheric dependence of absolute and percentage differences for $R_{rs}^{DD}(\lambda) - R_{rs}^{DD2}(\lambda)$ is shown in Figure 10. All four bands have the general trend that $R_{rs}^{DD}(\lambda) - R_{rs}^{DD2}(\lambda)$ increases with IDR. This relationship is strongly apparent in the 400 nm band, where in clear conditions the median difference values are close to zero, and in the most overcast conditions the median difference is >30%.

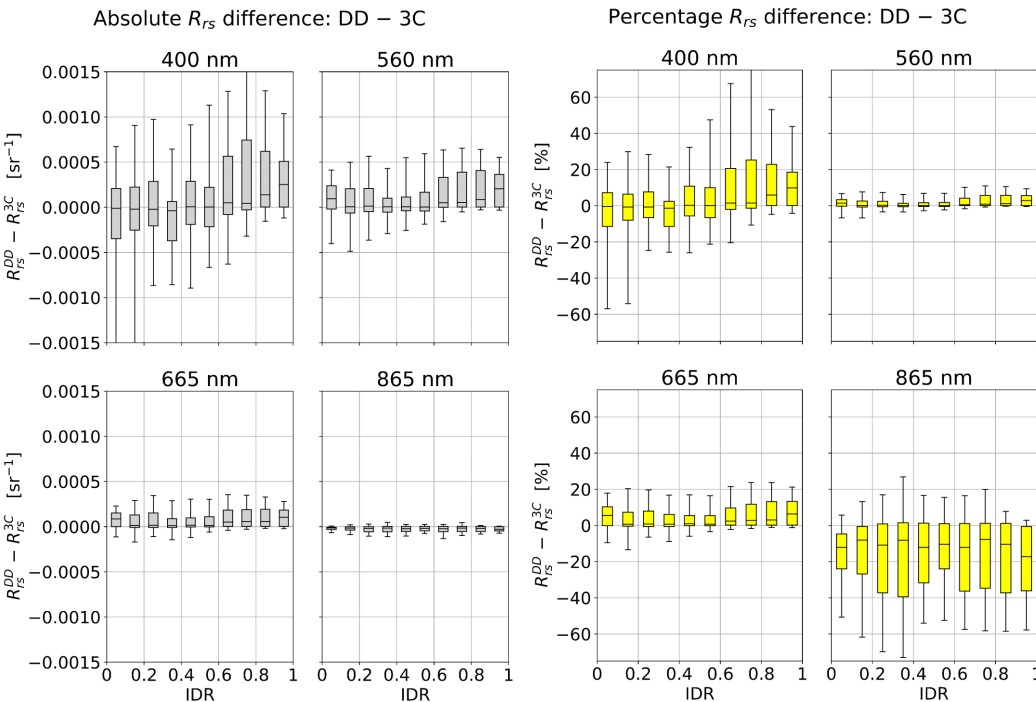

**Figure 9.** Atmospheric dependence of $R_{rs}(\lambda)$ differences for $R_{rs}^{DD}(\lambda) - R_{rs}^{3C}(\lambda)$. Left two columns: Absolute differences. Right two columns: Percentage differences defined as $200(R_{rs}^{DD} - R_{rs}^{3C}(\lambda))/(R_{rs}^{DD}(\lambda) + R_{rs}^{3C}(\lambda))$. The upper and lower box limits notate upper and lower quartiles, the central line indicates the median values, and the whiskers correspond to the 5th and 95th percentiles.

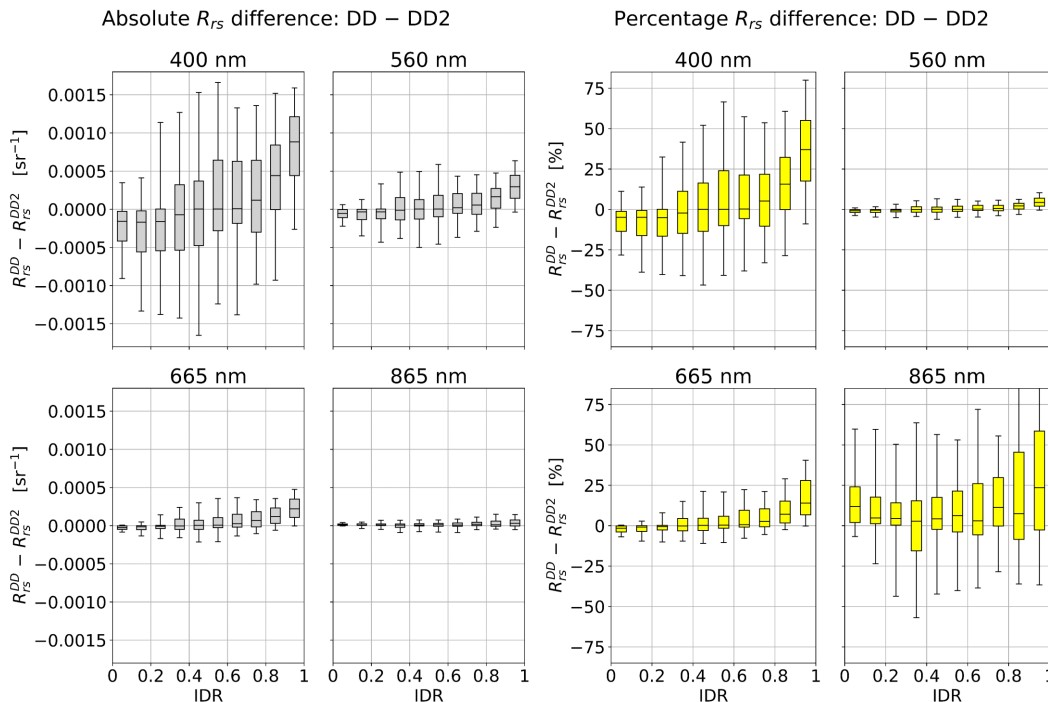

**Figure 10.** Atmospheric dependence of $R_{rs}(\lambda)$ differences for $R_{rs}^{DD}(\lambda) - R_{rs}^{DD2}(\lambda)$. Left two columns: Absolute differences. Right two columns: Percentage differences defined as $200(R_{rs}^{DD}(\lambda) - R_{rs}^{DD2}(\lambda))/(R_{rs}^{DD}(\lambda) + R_{rs}^{DD2}(\lambda))$. The upper and lower box limits notate upper and lower quartiles, the central line indicates the median values, and the whiskers correspond to the 5th and 95th percentiles.

### 4.4. Atmospheric Dependence of Glint Corrections and Algorithm Residuals

In general, the magnitude of $R_g(\lambda)$ increases with IDR. Therefore, to standardize the IDR dependence of the glint corrections across all conditions, we consider the glint-component fractions $R_s(\lambda)/R_g(\lambda)$, $R_{dd}(\lambda)/R_g(\lambda)$, and $R_{ds}(\lambda)/R_g(\lambda)$. These are shown for the three algorithms in the four spectral bands in Figure 11. Across all four bands, 3C and DD have an increase in $R_s(\lambda)/R_g(\lambda)$ with IDR. $R_{dd}(\lambda)/R_g(\lambda)$ is very small (<1%) for 3C across all IDR values. $R_{dd}(\lambda)/R_g(\lambda)$ and $R_{ds}(\lambda)/R_g(\lambda)$ are of comparable magnitude in clear conditions for DD, but in overcast conditions, the contribution of $R_{dd}(\lambda)/R_g(\lambda)$ is negligible. In clear conditions, both $R_{dd}(\lambda)$ and $R_{ds}(\lambda)$ are used in the DD2 glint correction, whereas in overcast conditions, only $R_{ds}(\lambda)$ is used. We remind the reader that $R_s(\lambda) = 0$ is set for DD2.

The atmospheric dependence of the algorithm residual $\epsilon$, Equation (11), is shown in Figure 12. DD and 3C have similar $\epsilon$ dependence on the IDR whilst DD2 has increasing $\epsilon$ with the IDR. The overall mean values of $\epsilon$ are 1.91 $\times 10^{-5}$ sr$^{-2}$ for 3C, 2.00 $\times 10^{-5}$ sr$^{-2}$ for DD, and 2.51 $\times 10^{-5}$ sr$^{-2}$ for DD2. It is noted that the modelled radiance ratios in 3C can be smoother than the measured ratios in DD and DD2 (e.g., Figure 6) which potentially impacts interpretation of $\epsilon$ in terms of retrieval accuracy. The increase in DD2 with IDR is likely to occur due to reliance on a single glint component ($R_{ds}(\lambda)$) in overcast conditions (Figure 11).

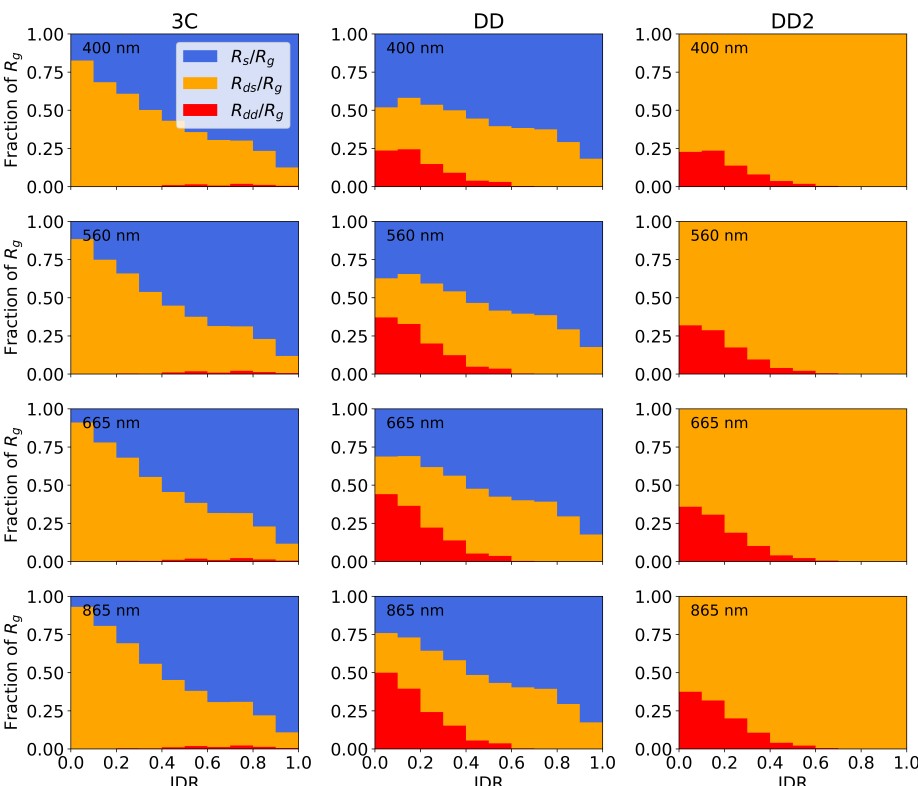

**Figure 11.** Dependence of optimized glint components (mean fractional value in each IDR bin) on atmospheric conditions. Rows: spectral bands; columns: algorithm variants.

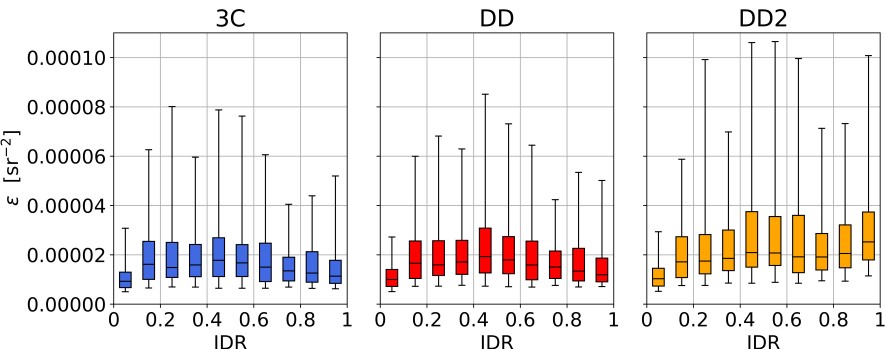

**Figure 12.** Dependence of algorithm residual on atmospheric conditions (IDR). The upper and lower box limits notate upper and lower quartiles, the central line indicates the median values, and the whiskers correspond to the 5th and 95th percentiles.

## 5. Discussion

Applications of in situ aquatic $R_{rs}(\lambda)$ include satellite validation (clearer atmospheric conditions only) and high-frequency monitoring of water bodies (extending across a wider range of atmospheric conditions). In situ $R_{rs}(\lambda)$ match-ups for satellite validation require strict adherence to fiducial reference standards [1,2,33]. High-frequency monitoring is generally less strict in requirements for $R_{rs}(\lambda)$, with increased utilization of data in sub-optimal conditions (e.g., higher wind speed) being a trade-off with data quality [11]. Here, we did not assess the absolute accuracy of $R_{rs}(\lambda)$ (proximity to a true reference value) but focused on precision (measured by $CV[R_{rs}(\lambda)]$). High precision does not ensure high accuracy, as there may be a bias present, but it is necessary for high accuracy. Past studies have shown the accuracy of 3C to compare favourably against established $R_{rs}(\lambda)$ schemes (e.g., [6,10]) in scattered cloud or overcast conditions [15,16], and the algorithm comparison in this study should be viewed in this context.

Comparative characterization of $CV[R_{rs}(\lambda)]$ as a function of atmospheric state (IDR: Equation (5)) was a central aim of this study (Figure 8). All algorithms have a sharp increase in $CV[R_{rs}(\lambda)]$ (decrease in precision) for IDR > 0.2 relative to IDR < 0.2. This is significant as it shows that clear conditions, as would be used in satellite validation, enable the highest-precision $R_{rs}(\lambda)$ estimates. It also shows that if satellite validation were performed in slightly non-clear conditions, there would be a decline in precision of $R_{rs}(\lambda)$. DD and 3C both have the highest $CV[R_{rs}(\lambda)]$ (lowest precision) for IDR in the range 0.4–0.6 (Figure 8) where scattered cloud and higher IDR variability is present (Figure 7). Therefore, filtering out these observations is a way to increase the overall precision of $R_{rs}(\lambda)$. It is noted, however, that a combination of wind-roughening [6,11] and asymmetry in the radiance distribution associated with scattered cloud will be two separate mechanisms that contribute toward higher $CV[R_{rs}(\lambda)]$. Therefore, partitioning the $R_{rs}(\lambda)$ variance due to IDR (or a similar atmospheric metric) and wind speed is a desirable future investigation. In overcast conditions, which have lower IDR variability than intermediate conditions (Figure 7), $CV[R_{rs}(\lambda)]$ is lower.

DD processing is designed to remove atmospheric model dependency from 3C by replacing model-optimized glint corrections, $(E_{dd}(\lambda)/E_d(\lambda))^m$ and $(E_{ds}(\lambda)/E_d(\lambda))^m$, with glint corrections measured by the HSP sensor, $E_{dd}(\lambda)/E_d(\lambda)$ and $E_{ds}(\lambda)/E_d(\lambda)$. This results in DD reducing the number of free parameters in the spectral optimization (specifically, dependence on the Angström exponent and turbidity coefficient). As the spectral shape of the glint correction in DD is better constrained by data than 3C, it is a plausible hypothesis that it results in more reliable $R_{rs}(\lambda)$. The result that best supports this hypothesis is reduced $CV[R_{rs}(\lambda)]$ (higher precision) of DD relative to 3C (Figure 8), which is observed in the blue part of the spectrum in clearer conditions. This is the spectral–atmospheric regime where the spectral curvature of the measured glint corrections is greatest (Figure 4) and is therefore consistent with DD having the curvature of the glint correction being better constrained within the optimization. Demonstration of improved algorithm precision in clear conditions is particularly important, as these are the conditions when satellite validation can be performed. Without independent $R_{rs}(\lambda)$ reference, the $R_{rs}(\lambda)$ differences between DD and 3C (Figure 9) cannot be used to definitively assess $R_{rs}(\lambda)$ accuracy. However, it is notable that the magnitude of $R_{rs}^{DD}(\lambda) - R_{rs}^{3C}(\lambda)$ is greatest in overcast conditions, which is when the atmospheric model in 3C [20] will be the worst match for the observed atmospheric conditions.

Differences in the optimized glint component fractions (Figure 11) provide insight into why 3C and DD produce different $R_{rs}(\lambda)$ and $CV[R_{rs}(\lambda)]$. In clear-sky conditions, 3C is heavily reliant on $R_{ds}(\lambda)$, which has its spectral shape determined by model-optimized $(E_{ds}(\lambda)/E_d(\lambda))^m$, whereas DD uses a mixture of all three glint components. Due to reliance on modelled glint, we speculate that 3C has a tendency to be overfitting in clearer conditions, which, in turn, could be the origin of the higher $CV[R_{rs}(\lambda)]$ at blue wavelengths (Figure 8). It is, however, noted that 3C $R_{dd}(\lambda)$ can be higher for glint-affected data [11], and that the relative contribution of $R_s(\lambda)$ has been higher in application of 3C to a different water type (Baltic sea data) [15]. In overcast skies, both 3C and DD become more reliant on the specular glint component, $R_s(\lambda)$. An explanation is that in overcast skies, $L_s(\lambda)/E_d(\lambda)$ and $L_t(\lambda)/E_d(\lambda)$ are of similar spectral roughness (e.g., the higher-frequency features in the NIR in Figure 6), which enables a better optimization fit than the smoother $E_{ds}(\lambda)/E_d(\lambda)$ and $E_{dd}(\lambda)/E_d(\lambda)$ curves. Physically, the spectral roughness occurs as radiance from a given direction will have travelled a different average atmospheric path length than irradiance; i.e., the spectral absorption imprints differ and do not cancel out as in clear conditions.

The two-sensor configuration of DD2 ($L_t(\lambda)$ spectroradiometer and HSP) is attractive as it enables equipment cost to be reduced from the three-sensor configuration of DD. The comparable $CV[R_{rs}(\lambda)]$ (Figure 8) and comparable difference $R_{rs}^{DD}(\lambda) - R_{rs}^{DD2}(\lambda)$ (where comparable reflects that the difference distribution is approximately centred about zero) in clearer skies (Figure 9) supports that, in these conditions, DD2 can well replicate DD $R_{rs}(\lambda)$ processing. An explanation for similarity in $R_{rs}(\lambda)$ is that $E_{ds}(\lambda)/E_d(\lambda)$ and

$L_s(\lambda)/E_d(\lambda)$ have similar spectral curvature in clear conditions (Figure 4). In intermediate and overcast conditions, $CV[R_{rs}(\lambda)]$ is significantly higher for DD2, and $R_{rs}^{DD}(\lambda) - R_{rs}^{DD2}(\lambda)$ increases with IDR. DD2 is solely reliant on the $R_{ds}(\lambda)$ glint component in overcast conditions (Figure 11), which is associated with an increase in the algorithm residuals (Figure 12) and likely (but not demonstrated) decline in $R_{rs}(\lambda)$ accuracy. It is therefore recommended to filter by IDR value if using DD2 in the future. A two-sensor system was previously explored for 3C processing using an $L_t(\lambda)$ sensor and a conventional $E_d(\lambda)$ sensor by Groetsch et al. [11] (which corresponds to the case where all of the glint correction basis functions are model-optimized). It was shown that spectrally-averaged $CV[R_{rs}(\lambda)]$ was lower for the two-sensor system than conventional 3C processing.

Previous implementations of 3C have considered both fixed [15,19] and variable [15,19] CDOM absorption slope ($S_y$) in the spectral optimization. The CDOM absorption slope has similar spectral dependence to $E_{ds}(\lambda)/E_d(\lambda)$, and a suitable parameterization is therefore believed to be key in separating atmospheric and water components in the spectral optimization [15]. In this study, which considered variable $S_y$, there was a slight tendency for termination at the upper bound ($S_y = 0.020$), but as this happened for <30% retrievals (averaged over algorithm variants), this is unlikely to have a dominant impact on interpretation our results. A hyperbolic decay function for CDOM absorption [34] can also be used in place of a conventional exponential decay for the CDOM slope in 3C processing [19]. This was experimented with but was not found to give a clear advantage for the Western Channel dataset (in the respect that a similar percentage of data overall reaches CDOM parameter bounds, regardless of slope model). A scalar offset, $\delta$, can also be added to the definition of 3C $R_{rs}(\lambda)$ in Equation (3) [19]. We considered this as a control run but the overall results were very similar (e.g., median percentage $R_{rs}(\lambda)$ differences were typically within ~1% of values in Figures 9 and 10).

## 6. Summary

In this investigation, we incorporated a novel hyperspectral pyranometer (HSP) [24] within above-water $R_{rs}(\lambda)$ processing from a solar-tracking radiometry platform (So-Rad) [23]. HSP measurements of direct and diffuse irradiance enabled us to adapt the 3C (three-component glint) $R_{rs}(\lambda)$ algorithm to remove atmospheric model dependency of a spectral optimization procedure, which we termed the DD (direct-diffuse) algorithm. The HSP measurements then enabled a comparative investigation of $R_{rs}(\lambda)$ as a function of atmospheric optical state, including benchmarking against a two-sensor version of DD (DD2).

Our investigation showed that 3C and DD have similar $R_{rs}(\lambda)$ precision (measured via the coefficient of variation over a 20 min measurement cycle) at green, red, and NIR wavelengths. DD has improved precision over 3C at blue wavelengths in clear-sky conditions, which marks a key advantage of using the HSP sensor in $R_{rs}(\lambda)$ processing. Improved algorithm precision in clear conditions is particularly desirable, as these are the conditions when satellite validation can be performed.

As it reduces the number of free parameters within the algorithm spectral optimization, and has the spectral shape of the glint correction based purely on measurements, it is anticipated that DD produces more accurate $R_{rs}(\lambda)$ than 3C. Future work should aim to assess DD $R_{rs}(\lambda)$ accuracy across a range of optical water types against in-water reference reflectance measurements.

The HSP is also a standalone sensor for aerosol characterization [24,25] and can therefore be considered for range of atmospheric monitoring applications. Within the context of applications of *in sit* $R_{rs}(\lambda)$, the additional characterization of atmospheric properties provides valuable information on sources of uncertainties in atmospheric correction of satellite observations.

**Author Contributions:** Conceptualization, S.G.H.S. and P.M.M.G.; methodology, T.M.J., S.G.H.S., P.M.M.G. and J.W.; software, P.M.M.G.; investigation, T.M.J., S.G.H.S., P.M.M.G. and J.W.; resources, S.G.H.S. and J.W.; writing—original draft preparation, T.M.J.; writing—review and editing, T.M.J., S.G.H.S., P.M.M.G. and J.W.; project administration, S.G.H.S.; funding acquisition, S.G.H.S. All authors have read and agreed to the published version of the manuscript.

**Funding:** This project has received funding from the European Union's Horizon 2020 research and innovation programme under grant agreement No 776480 (MONOCLE: Multiscale Observation Networks for Optical monitoring of Coastal waters, Lakes and Estuaries), https://monocle-h2020.eu (accessed on 10 May 2022).

**Data Availability Statement:** Upon publication, new reflectance algorithm codes will be available from an open-source repository. The 3C algorithm code is available at https://gitlab.com/pgroetsch/rrs_model_3C (accessed on 10 May 2022) and https://gitlab.com/pgroetsch/rrs_model_3C/-/tree/hsp (accessed on 10 May 2022). So-Rad data and quality control are available at https://pypi.org/project/monda/ (accessed on 10 May 2022).

**Acknowledgments:** The authors would like to thank Thomas Jackson and Adam Wright, both at Plymouth Marine Laboratory, for discussions on data processing and development of the So-Rad system, respectively. We would like to thank Alexandre Castagna, University of Ghent, for his helpful comments on an earlier version of the manuscript. We are also grateful to Brittany Ferries and, in particular, the crew of the *Armorique* for providing access to their vessel.

**Conflicts of Interest:** The authors declare no scientific conflicts of interest. The So-Rad sensor platform was produced by and can be purchased from Plymouth Marine Laboratory. Construction guides and software are freely available for non-commercial use. The HSP-1 sensor is developed by Peak Design Ltd., and is expected to become commercially available in the near future. Philipp M. M. Groetsch is employed by Gybe.

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
