# Peer review of "Incorporating a Hyperspectral Direct-Diffuse Pyranometer in an Above-Water Reflectance Algorithm"

_remotesensing, doi:10.3390/rs14102491_

Round 1

Reviewer 1 Report

Review of "Incorporating a hyperspectral direct-diffuse pyranometer in autonomous above-water reflectance processing" by T. M. Jordan et al.

This manuscript proposed a novel algorithm for estimating remote sensing reflectance (Rrs) using data observed by a hyperspectral pyranometer (HSP) and compared with other algorithms. The algorithm refered as DD in the manuscript is practical in continuous observations and is also effective in validating satellite data. It is an algorithm that can be expected in the future. I think this manuscript is worth publishing in this journal, but some questions remain before that.Comments are as follows.

Major comments

- I think it's better to reconsider the title. The novel algorithms obtained as a result of using HSP, but the content of the manuscript is mainly comparison and verification of new algorithms.

- Althogh authors wrote "DD has improved precision over 3C at blue wavelengths in clear-sky conditions,..." (l.519-520), it is difficult to understand whether DD was "imploved" from the comparison of CV and the difference between Rrs for each algorithm. Please add an explanation whether CV will lead to precision.

Minor comments

- Equation 3 and 4
  Is the dimension on both sides equal?
  (especially Equation 4)

- 2.1 Field data
  -- Some people are not familiar with Armorique, so it would be better to write gross tonnage.
  -- How was the observation time? Did the observation perform 24 hours a day? What time to what time did you use it for processing? It is better to describe these things.

- Figure 1
  It is better to describe how the RRS in the figure is calculated.

2.2 The solar-tracking radiometry platform (So-Rad)

- Figure 2a
  The position of the Ed sensor seems to be lower than the Ls and Lt sensors. I think this is wrong design for an Ed measurement. Even if the effect from the Ls and Lt sensors is small, the position of the Ed sensor must be located above the Ls and Lt sensors. This study did not use the ED sensor. However, as there is this photo, I think it is better to show the positional relationship between the HSP sensor and the Ls and Lt sensors.

- Figure 3
  (It may be my misunderstanding.)
  Why is the Eds of the overcast higher than the Eds of the clear conditions?

- l.227-228
  The position of "m" is different from Equation 4.

- l.396
  Since the original value is small, it is natural that the percentage increases with a small difference.

- l.415
  The position in Figure 11 is too far from the position of the text.

Reviewer 2 Report

In this manuscript, the authors present a new method for autonomous above-water reflectance processing by creatively incorporating a hyperspectral direct-diffuse pyranometer in the spectra collection. Using this method, the direct and diffuse irradiance can be measured with the pyranometer and the processed Rrs has lower variability than the 3C model in the blue part of the spectrum in clearer sky conditions. Some comments are listed below:

  1. It is important to cross-validate the sensors for the total upwelling radiance Lt, sky radiance Ls, and downwelling irradiance Ed in the Rrs processing. How was this done with the pyranometer?
  2. Line 122 – 124. Am I right that only three stations’ data were used in the following analysis?
  3. Figure 4 and Figure 5. Are the two figures illustrating that the Eds/πEd and Edd/πEd modeled using the 3C model are not accurate at all? How much difference will be transferred to the processed Rrs?
  4. Line 322-325. As you said the Rds and Rdd can be small compared to Rs, how to neglect the Rs in the DD2 method?
  5. Although the ship is stationary, the water can be changing over time. How the authors consider the change of water optical properties in the variability analysis?
  6. Line 376-382. What is the reason behind the atmospheric dependence of Rrs variability using different methods?
  7. Line 387. I think it should be “the aim is to establish…”
  8. Section 4.3. The reason for the atmospheric dependence of Rrs differences with different methods also should be explained.
  9. Line 428-431. Again, how this precision was quantified while the water may be changing over time?
  10. Line 450-463. I understand the authors tried to explain the reason for the better performance of DD, but the strengths of the new DD method compared to 3C model should be delivered clearer.
  11. Line 481-493. I doubt the DD2 method can achieve comparable results of Rrs with the DD or 3C method in clear sky, when it neglects the sky radiance.

Round 2

Reviewer 1 Report

I think that it is true as a physical explanation about Figure 3 answers. However, a question still remains.
The question is that the Eds spectrums in the clear conditions and overcast conditions, which are shown in the bottom row of Figure 3, are quite similar. It is natural that the blue color is strong in the clear conditions.
On the other hand, Eds in overcast conditions should show a more flat spectrum. For the above reasons, the spectrum in the bottom row in Figure 3 looks so unnatural. However, this point is not related to the content of this manuscript, so there is no need to correct it.

I think it is better to correct only one of the following points.

・ The caption of the table is described on the table.

I think that the other points have been sufficiently modified.
I look forward to future developments.

Author Response

Second round of comments to reviewer 1

Again, we would like the reviewer for their positive and constructive comments. Our feedback is in italics.

Comments and Suggestions for Authors

I think that it is true as a physical explanation about Figure 3 answers. However, a question still remains.
The question is that the Eds spectrums in the clear conditions and overcast conditions, which are shown in the bottom row of Figure 3, are quite similar. It is natural that the blue color is strong in the clear conditions. On the other hand, Eds in overcast conditions should show a more flat spectrum. For the above reasons, the spectrum in the bottom row in Figure 3 looks so unnatural. However, this point is not related to the content of this manuscript, so there is no need to correct it.

The data in Figure 3 (overcast conditions) is correct and is broadly representative of overcast Eds and Ed spectra that occurred throughout the deployment. In general, the spectral shape of Eds and Ed is consistent across conditions, because the scattering by the clouds is approximately spectrally neutral across these wavelengths.

Figure 2 in Simis and Ollson 2014 shows a good example of the shape of Ed being similar across all conditions (and it will follow that Eds will be too), so we have added: The spectral shape of $E_{d}(\lambda)$ is generally similar across all atmospheric conditions \cite{Simis2013}. (line 216 in new submission).

We would also like to highlight the changing y-axis scale between the plots

I think it is better to correct only one of the following points.

The caption of the table is described on the table.

We are not clear about what is meant by this comment. Our caption is inserted outside of the `tabular’ environment in Latex, and the PDF formatting looks ok to us (caption beneath table). If there is an issue, we will correct it with the MDPI publishing team when the final typesetting is formatted.

I think that the other points have been sufficiently modified.
I look forward to future developments.

Thanks again for your positive feedback.

Extra changes 

Following reviewer 1’s feedback from the last review, we did a final formatting of figure size and positioning in the text.

Some minor typographical errors were spotted on our proofread of the revision and were corrected. There were no changes to the scientific content.

Reviewer 2 Report

The authors have answered all of my questions.

Author Response

We would again like to thank the reviewer for their constructive comments in the first review that improved our MS.